# Clinicians’ and Researchers’ Views on Precision Medicine in Chronic Inflammation: Practices, Benefits and Challenges

**DOI:** 10.3390/jpm12040574

**Published:** 2022-04-03

**Authors:** Anke Erdmann, Christoph Rehmann-Sutter, Claudia Bozzaro

**Affiliations:** 1Institute for Experimental Medicine, Medical Ethics Working Group, Kiel University, 24105 Kiel, Germany; claudia.bozzaro@iem.uni-kiel.de; 2Institute for History of Medicine and Science Studies, University of Lübeck, 23552 Lübeck, Germany; christoph.rehmannsutter@uni-luebeck.de

**Keywords:** precision medicine, personalized medicine, justice, data misuse, discrimination, ethnic group, access, shared decision making, doctor–patient relationship, broad consent

## Abstract

(1) Background: Due to the high burden of diseases with chronic inflammation as an underlying condition, great expectations are placed in the development of precision medicine (PM). Our research explores the benefits and possible risks of this development from the perspective of clinicians and researchers in the field. We have asked these professionals about the current state of their research and their expectations, concerns, values and attitudes regarding PM. (2) Methods: Following a grounded theory approach, we conducted qualitative interviews with 17 clinicians and researchers. For respondent validation, we discussed the findings with the participants in a validation workshop. (3) Results: Professionals expect multiple benefits from PM in chronic inflammation. They provided their concepts of professionals’ and patients’ work in the development of PM in chronic inflammatory diseases. Ethical, process-related and economic challenges were raised regarding the lack of integration of data from minority groups, the risk of data misuse and discrimination, the potential risk of no therapy being available for small strata, the lack of professional support and political measures in developing a healthy lifestyle, the problem of difficult access to the inflammation clinic for some populations and the difficulty of financing PM for all. (4) Conclusions: In the further research, development and implementation of PM, these ethical challenges need to be adequately addressed.

## 1. Introduction

One of the greatest medical achievements of recent decades has been the discovery that inflammatory processes are associated with a wide range of chronic diseases. Indeed, chronic inflammation-related diseases such as ischemic heart disease, diabetes mellitus, stroke, cancer, chronic kidney disease, non-alcoholic fatty liver disease, autoimmune and neurodegenerative diseases are responsible for more than 50% of all deaths worldwide [1]. The adoption of a western lifestyle in newly industrialized countries has increased the incidence of inflammatory diseases, as the example of inflammatory bowel disease (IBD) shows [1,2,3,4]. For example, the proportion of people with IBD in China has increased parallel to the “westernization of diet and culture” [5]. This development exemplifies the importance of lifestyle and environmental influences in the development of inflammatory diseases and supports the thesis that heritable factors are less important than primarily assumed [1]. But in addition to these factors, chronic infections, physical inactivity, microbiome dysbiosis, dietary factors, environmental and industrial toxicants, along with social and cultural changes are involved in the etiology of chronic inflammation [1]. The accumulation of these influences appears to cause an increase of chronic inflammation in many populations.

Due to the high burden of diseases with chronic inflammation as an underlying condition, great expectations are placed in the development of precision medicine (PM). The goal of PM is to provide the right therapy to the right patient at the right time, while increasing the therapy response and reducing side effects [6]. With deep phenotyping and artificial intelligence, a large amount of individual data can be processed for each patient: data on health status, nutrition, genetics and epigenetics, microbiome, metabolomics, proteomics, physical activity, stress, sleep, age, geography, socioeconomic status and early life [7]. These “multi-omics data” [8] are required to finally stratify patients into subgroups that differ due to individual characteristics and to the likelihood of a positive therapy response [9].

However, although PM promises to be highly beneficial, and some knowledge about ethical challenges that need to be addressed is available, little is known about researchers’ and physicians’ own views on the research procedures that make PM in chronic inflammation special and the ethical questions it raises for them. Some of the ethical challenges have already been described in the bioethical literature [10,11,12,13] and, similarly, the perspectives of professionals and patients regarding ethical issues in different medical fields have already been investigated. A recent review synthesizes patients’ and professionals’ views related to ethical issues in PM from 92 empirical studies conducted in different medical fields [14]. According to this review, many professionals have a positive attitude towards PM in general. They associate it with high benefits, but also identify risks. For example, professionals are concerned about the lack of evidence for the accuracy of diagnostic tests or the efficacy of therapies. They perceive patients as having limited knowledge about PM, which makes obtaining informed consent difficult. They expect limitations in access to PM for underprivileged people and ethnic minorities. Other problems they mention is the possibility of data misuse by insurance companies and employers, the potential of racial stigmatization due to genetic characteristics or the unwanted communication of incidental findings. Some professionals can imagine a change in the doctor–patient relationship if the physician focuses primarily on data and places less importance on talking with the patient. They also express concern about whether health insurance companies will cover the costs [14].

Most of the studies included in this review were conducted in oncology, an area in which PM is well established and positive therapy responses have already been achieved through PM, for example in gastric cancer [15]. Professionals working in the treatment of inflammatory diseases or conducting research in this area were less frequently interviewed [14]. Therefore, we decided to conduct a study on the perspectives of clinicians and researchers in the field of inflammatory medicine regarding the work procedures and the ethical issues in PM. For this study, we developed the following research questions:What are expectations, concerns, values and attitudes of experts regarding PM in chronic inflammatory diseases (CID)? What do they perceive as benefits and risks?How does PM improve the care of patients with chronic inflammatory diseases? In which way are patients and their families burdened by PM?What is the impact of PM on clinical (diagnostic and therapeutic) decision making and how can decision making be improved?How does PM influence communication between clinicians and patients? How can communication be improved?What is a “good life” with CID and how can PM contribute to a good life with CID?

Our study is part of a larger project in which we are investigating the potentials and limitations of PM in chronic inflammatory diseases from the perspectives of researchers, clinicians and patients. One of the goals of this project is to develop recommendations for researchers, physicians and policymakers on how clinical decision making and communication can be improved and how future health policy regarding PM can be shaped. Since a particular focus in this project will be on the question of to what extent PM can contribute to a good life with CID, an additional aim of the project is to refine or develop new patient-reported outcomes (PRO) for PM. These PROs can later be used to measure PM’s contribution to a good life with CID. 

## 2. Materials and Methods

Between January and August 2021, we recruited (via email) 17 experts who were (and still are) working in the field of chronic inflammation at a university hospital in Germany. The experts were: clinicians (physicians, study nurse), researchers (biologists, informaticians, bio-informaticians, geneticists, interface designer) or clinician scientists who worked both in the clinic and the laboratory (Table 1). We asked the experts to participate in a qualitative interview after informing them with an information leaflet about the aims, benefits, possible risks and the study procedure. All participants provided written informed consent. Due to the COVID-19 pandemic, we mainly conducted the interviews via video chats or phone calls, with only two face-to-face interviews. We used an interview guide based on our literature review [14] to ensure that all topics of interest were addressed. The interviews lasted between 32 and 93 min and were recorded with an audio recorder and transcribed verbatim. Before starting, we obtained approval from the ethics committee of Kiel University.

For data collection and analysis, we chose the grounded theory approach according to Strauss and Corbin. This approach aims to develop an inductively generated, data-based theory about a phenomenon [16], in this case PM in CID and its ethical challenges. The research style of grounded theory has the following characteristics. 

First, data collection and analysis alternate continuously. After an interview has been conducted and transcribed, it is immediately open-coded before the next interview starts. In this way, the researcher’s preconception is modified right after the first interview and this new conception is fed into the next interview [16]. 

Second, this modified preconception is also the basis for the selection of the next study participants. This process is called theoretical sampling: participants are included in the study who are expected to further differentiate the emerging theory. If new participants do not contribute any more new findings, the data are considered saturated and the data collection can be finished [16]. 

Third, the interviews are coded in three steps: in open coding, the data are read line by line and the phenomena enclosed are labelled with codes. The codes are grouped and characteristics or dimensions are identified (for an example, see Table 2). During axial coding, the categories developed in this way are related to each other. According to the grounded theory coding paradigm, in this step, the following is worked out: conditions that cause the phenomenon, the context in which the phenomenon is embedded, the interactions and strategies of the actors involved and the consequences of these strategies. Finally, through selective coding, the central theme of the story is elaborated by identifying a core category that is related to all other categories [16]. Open coding and axial coding were performed in MAXQDA, a software program for qualitative data analysis.

We adopt a constructivist position in our research, which means that we understand our perceptions in the research process as cognitive constructions and that these constructions do not necessarily correspond exactly with reality. Rather, we assume that our perceptions at the same time blend out other possible perceptions and recognize that we can never fully grasp objective reality [17]. However, in order to overcome this researcher bias as far as possible, we carried out member-checking [18] after completing and analyzing the interviews. For this purpose, we invited all interview partners to a validation workshop where we presented the key findings and gave the participants the opportunity to comment on them. We also made the results available in a written form before the workshop and informed the participants about the questions that should be clarified during it:Is your opinion adequately represented in the results?In your view, are there any misinterpretations of the data?What should be added to the results?Do you have difficulties in understanding, or any questions?

Four experts participated in this workshop; they gave us valuable information that we added to our interpretation. 

## 3. Results

From open coding, six main categories emerged: *inflammatory diseases, precision medicine, professionals’ work, patients’ work, problems inherent in the system* and *ethical challenges*. The core category of our grounded theory, to which all other categories are related, is *professionals’ work*. This category contains all of the information describing what professionals do, or intend to do, in order to develop, apply and evaluate PM in CID. We describe this category in the following section and in the subsections, we report about the other categories that represent in summary different challenges related to PM.

### 3.1. Work of Professionals in the Development of PM in CID

In the development of precision medicine, the clinicians and researchers are currently working on the development of a valid database consisting of different patient data. With multi-omics data and by using machine learning, they intend to discover biopatterns that allow patients to be stratified into groups. The aim of these efforts is to offer patients a more precise therapy or prevention concept that is tailored to their individual data. However, almost all respondents state that this new type of medicine is not yet being applied, as more data are needed to discover biopatterns (Figure 1). 

The development and application of PM in CID requires interdisciplinary collaboration between different disciplines (Figure 1). Some respondents explain this by the fact that these diseases are systemic diseases that usually affect not only one organ, but also further organs in the course of their progression. This knowledge prompted the responsible experts to establish a molecular inflammation board in which various experts come together to discuss therapy.

“On the other hand, this is simply due to the fact that with chronic inflammatory bowel diseases, with rheumatoid arthritis, we are practically dealing with inflammatory systemic diseases. And they usually affect one organ primarily. In the course of the disease, however, there is also secondary involvement, other organ systems are affected. And then it becomes complex and you can no longer assess it on your own. Then you need expertise and this inflammation board was founded to bring this expertise together.” (Interview 55, clinician) (Please note that we conducted the interviews in the German language and translated the experts’ remarks into English for this paper.)

#### 3.1.1. Challenges in Developing a Valid Database

In developing a valid database for PM, experts report some challenges that need to be addressed. For example, one expert explained that access to, or exchange of, patient data for research is difficult due to inconsistent data processing systems in German hospitals. This researcher suggested that the exchange of data between hospitals should be improved by a uniform, nationwide IT infrastructure. In addition, informed consent should be standardized among hospitals:

“The second point is, as mentioned, both the patient informed consent form, patient information: this broad consent that is needed in order to use data. It is not uniform either, and in some cases, it is incomplete, so that it is not always known which patient information sheets I can use, which I am not allowed to use, and how I can use them for research.”(Interview 63, researcher)

Broad consent (BC), which allows researchers to use collected data and biospecimens for future research without repeated patient consent, is viewed positively by the majority of respondents (Table 3).

However, BC also has disadvantages, which some respondents described. One clinician pointed out the lack of standardization of the samples obtained with BC and emphasized that not all samples are eligible to answer every research question:

Another respondent took the patient perspective and described concerns about BC. This person would prefer a dynamic consent, which offers a new opportunity to consent to a new study.

“Personally, I’m not entirely a fan of it [BC], at least not in the sense that I’m not further informed about where this data actually ends up. I would like to be informed about which studies my data go to and how they are used. In the course of my [work], I stumbled across dynamic consent and found the approach quite interesting, because there are also different approaches for platforms where the respondent can actually decide from study to study to which study, he or she provides the data and to which not. And I personally would like that much better, because then I would be more actively involved again, and not just consent once and then give a free pass for everything that comes after that.”(Interview 65, researcher)

In contrast, for one clinician, even BC is not sufficient and this person would like to have the possibility to analyze patient data retrospectively.

“So, I would like to see a broad consent that would really make it watertight again: a patient is treated in the [University clinic] and we are allowed to do everything with the data pseudonymously, but we are also allowed to read out the data ourselves, of course, that one reads the name sometimes, but that one has a low-threshold access to the data among each other. (…) For example, if I have a doctoral student who wants to retrospectively analyze all my patients who have received one of these new therapeutic agents, I would like to evaluate how they fared in routine care, how long they received this medication, how they responded. I would like to be able to put a doctoral student at a computer and have them evaluate the data. That’s not possible.”(Interview 62, clinician)

However, as the validation workshop clarifies, data protection in general is considered to be beneficial and essential. However, the inconsistent interpretations of the legal regulations by various supervisory authorities are criticized:

“That it is not a global criticism of data protection per se, which I believe is close to the heart of every medical researcher, but it is the heterogeneous interpretations of the legal requirements that sometimes make this very, very incalculable and difficult and problematic. If everyone would know exactly where to stand, then everything would be fine, then we would know exactly what we can do, what we can’t do, how we have to do it.”(Validation workshop, researcher)

Apart from the problems concerning the collection of data and biospecimens, an additional problem is the integration of data from wearables or smartphone apps into patients’ electronic health record (EHR) for which the technical interfaces have not yet been provided. The quality of data from these devices cannot yet be assured in the same way as, for example, the data from a laboratory, as one expert pointed out:

“At the same time, there is always the question of how the validity of this information is checked. So, if you give a laboratory sample—a blood sample—to the laboratory, then you know that if the hemoglobin value is so and so, then it is. I just rely on that now. That’s of course with all this information, it’s much more changeable.”(Interview 55, clinician)

Genetic information requires special protection and this is the reason why, in Germany, it is not integrated into the EHR. However, this poses a problem for the development of PM, as one expert described:

“What is very, very difficult is to transfer genetic data in this form back to the patient’s file, for example. (…) And here, for example, it is the case that these genetic findings are not actually allowed to appear electronically (…) in the patient information system, but are always communicated to the patient by post (…). And there is now the question of how we should make decisions with the help of genetic data. (…) And there I see a grey area, so to speak, at the moment, as to how exactly this is handled now.”(Interview 57, clinician)

An unsolved problem reported by some experts is also the fact that biomaterial for genome research is predominantly obtained from persons with European ancestry and may in future lead to disadvantages in PM for people with a different ancestry. This problem obviously results from a lack of research funds in many non-European or non-American countries and can mean that newly discovered biomarkers are of no use in other population groups. One respondent also recognized this problem in the German context:

“Three million, four million Turks, people of Turkish origin [living now in Germany], who are genetically so far removed from northern Germans or [other] people who have lived here for centuries. That could lead to such injustices that the markers are of no use.”(Interview 67, researcher)

In the validation workshop, one participant added that this problem not only plays a role in precision medicine research, but also in routine clinical research:

“But of course, this is true for any form of medical research. Also, for example, clinical research, because the transferability of research results always depends on the population where I apply it afterwards corresponding to the population where I did the research. So, for example, we have the problem that many pharmaceutical studies were carried out in collectives that had a completely different sex ratio than the typical patients who are supposed to benefit from them afterwards. And that we have, let’s say, a male pharmacology in many places.”(Validation workshop, researcher)

In addition, the experts are aware of the challenge that the extensive collection of data poses a risk of misuse, which could be relevant for the patient, but also for their children when genetic data are involved. One respondent pointed out that simply installing a health app on a smartphone carries the risk that patients will be identified as having a particular condition.

#### 3.1.2. Challenges in Patient Education and Therapeutic Decision Making

Since patients have different levels of health literacy, the clinicians pursue two strategies to provide them with adequate information: a) adapt to different levels of competence or b) generally provide simple explanations. Despite these efforts, some experts reported that patients sometimes feel overwhelmed by shared decision making (SDM). 

“Many people are actually surprised that we really ask so many questions, because, as I said, they are not used to it. Then there is also, for example, shared decision making. Patients are often a bit overwhelmed by the question, ‘What do you think about this? Well, we have now presented the therapy options here, so to speak, and what do you think? What would be feasible and comfortable for you?’ And then the questions often come up, “Yes, you are the doctor, you have to decide that.”(Interview 54, clinician)

This result corresponds with the experience of clinicians that many patients, especially older ones, have a high level of trust in their doctors. They trust that the doctors will do the right thing. This trust also seems to be necessary in the context of PM, as one expert explained. This person assumed that as a result of PM, shared decision making will not be possible anymore because the information required for PM is too complex to understand and patients are therefore forced to believe what doctors say.

“The risks are that the more precise and deeper you go, the more shared decision making will be undermined. Because in the end there will be no more shared decision making. (…) Because (…) things (…) become so complicated that the individual can no longer judge them comprehensively. They have to believe it, yes. That’s the first problem. The second is if you don’t understand molecular processes that lead to it, at some point it no longer works. [The patients] just have to believe it, like that.”(Interview 66, clinician)

One could therefore assume that the right strategy for PM is to ‘convince’ the patients, a word used by some respondents. However, the results from other interviews and from the validation workshop show that the underlying concept of SDM presented in the quote above is not supported by all participants. As another expert explained:

“And just because we don’t explain all the mechanisms of our therapy in detail, I can’t really see that shared decision making shouldn’t work in precision medicine either. In general, we have biomarkers that tell us whether a therapy responds better or not. And that’s how it can be said. “We have found some genetic aspect in your case, so we assume that therapy A doesn’t work so well and that’s why we want to use therapy B, as it is better tailored to you.” You can say that without explaining the genetic defect in detail.”(Validation workshop, clinician)

In addition, several interview excerpts show that the focus of education is primarily on what the recommended therapy means for the patient’s daily life: 

“And then we look together with the patient to see what is possible. That doesn’t mean what would be best for us as doctors, but what can the patient actually implement. So we ask, “Can you take any tablets at all?” Or, “Can you give yourself injections independently? What about work, occupation?” Because every therapy has possible side effects. And of course they should not interfere as little as possible with everyday life.”(Interview 54, clinician)

So there seem to be different views on SDM and these varying perspectives are also evident when it comes to defining therapy goals. While, as presented above, one expert believed that the doctor must set this goal, another expert pointed out that these therapy goals can be developed just as much with the patients.

#### 3.1.3. Impact of Healthcare and Research System

In addition to the question of shared decision making, certain system-inherent structures of the healthcare and research system also influence decisions for or against a therapy. For example, from the perspective of the experts, one problem in decision making is the lack of evidence for certain therapies. On the one hand, certain population groups are not included in clinical trials (the very young and the very old), and on the other hand, PM stratifies into small groups for which the statistical evidence is very low. In the worst case, one respondent stated it is possible that no therapy will be available for individual patients because their characteristics do not fit the algorithms developed within the framework of PM. In addition, if the strata are too small, it may no longer be worthwhile for pharmaceutical companies to invest in drug or marker development, as the following expert explains: 

“I don’t think any pharmaceutical company can be expected to spend millions on the development of a marker that can only be used once in a fraction of the population. So I believe that this personalization means under certain circumstances, personalization also means exclusion.”(Interview 67, researcher)

Finally, several experts claimed that in the future, capacities and costs will probably determine which therapy is possible for individual patients. At present, there are already some people who cannot be treated in a certain clinic for capacity reasons or be prescribed with certain drugs because they are not covered by insurance:

“For me, the biggest issue is that (…) some medicines are not paid for by the health insurance, although they are effective and also make absolute sense for the patient and are also, let’s say, the best from my point of view in terms of the risk for side effects. But they are not paid for. Or that I actually already know how I would prefer to treat my patients (…) but often there are rather step-by-step therapies in the guidelines, that one first does this and then that and then that, instead of rather carrying out a more efficient treatment from the beginning. These are the ethical problems I have. Yes, that I actually feel restricted in the drug therapies.”(Interview 48, clinician)

Further, another expert criticized the fact that practitioners have to take costs into account when choosing medicines:

“Actually, as a physician or as a nurse (…), we should not actually know what these drugs cost, so that we use them all in the way that makes medical sense. And not because they’re cheap or expensive and well, who knows, because they’re all ultimately licensed and the cost is something that other parties in the healthcare system have to talk about, but not those who are with the patient.”(Interview 62, clinician)

One respondent suggested that the severity of the disease could be a criterion for access to PM. Patients who are well cared for with standard medication could be maintained on these medications for cost reasons:

“Of course, it only makes sense to think about this kind of well, precision medicine, if the cases are more difficult. If someone is treated well, with a quite normal drug, cheaply, a cheap standard drug, then one will not have to think about it any further.”(Interview 57, clinician)

In conclusion, the financing of precision medicine is seen as an economic challenge and raises questions about the fair distribution of the resources:

“The question is of course, how is the whole thing financed. As I said, even for tumor patients, we can only afford it for a handful of patients at the moment. The question is, of course, who gets it and who doesn’t get it. You have to have distribution criteria. And at the end of the day, the therapy in question is perhaps very, very expensive, rather than taking the blockbuster out of the drawer. And even if we have great therapy proposals, that doesn’t mean that the therapies are available. Especially at acceptable prices. (…) So the whole thing is also an economic, an economic challenge.”(Interview 51, researcher)

#### 3.1.4. Patient Work as Challenge with Regard to Development of Precision Medicine

As is already the case in medicine today, the success of PM for CID will also depend on patient cooperation. For PM, patients have to provide data, for example, by filling in questionnaires, keeping a patient diary, using wearables and undergoing numerous examinations. As part of SDM, they have to make treatment decisions. In the future, patient cooperation will probably be even more important for therapy and prevention, as they have to work on their own health (Figure 2). This “patient work” [19] will be an essential part of PM, as one expert explained by giving an example:

“Like when if it turns out that due to a certain biopattern, a patient with rheumatoid arthritis has a particular susceptibility when their diet is low in omega 3. It is also possible that individualized results show that one person benefits better from a change in diet than another patient does.”(Interview 53, clinician)

However, such patient work also poses some challenges. Adherence to therapy is sometimes insufficient for various reasons, such as patients stopping medication when they feel better or not taking prescriptions at all because they cannot afford the co-payment. Sometimes the patients doubt the proposed therapy after they have returned home. One respondent noted that traditional treatment beliefs can interfere with treatment adherence in some migrant populations: 

“The mother comes and hands the patient some antibiotic, and they take the antibiotic, because she [the mother] was treated with it somewhere in Syria or somewhere else when she was a child.”(Interview 66, clinician)

When it comes to following a certain diet or losing weight, experts reported that patients are often overwhelmed with this task and feel guilty if they fail in their efforts. In order to make these tasks easier for patients, one respondent considered policy measures to be necessary:

“But we have also learned that the patient alone is completely overwhelmed with this problem. (…) Achieving weight loss at the broader level is not possible without political guidelines. So, this famous nutrition traffic light must come. A sugar tax on sweetened drinks, for example, must be introduced. Without these aids, the patient alone is not able to lose weight simply because they want to. It does not work.”(Interview 50, clinician)

Another expert reported that patients sometimes have unrealistic therapy goals. Therefore, this person highlighted the need for patients to receive psychological, nutritional and physiotherapeutic counselling in order to facilitate these tasks:

“That is the point I wanted to make with the psychologists. That in practice, precision medicine is also understood in such a way that perhaps individualized programs—psychologically, behaviorally or whatever—tailored to the patient are developed, so that therapy adherence can also be successfully maintained over a long period of time. Only when people get it into their heads that adherence to therapy is just as important as the actual diagnosis and initiation of therapy, will they realize that it is only possible with the patient. And then the patient will also come back to the center of treatment.”(Interview 53, clinician)

However, this clinician also problematized that the capacities of non-physician professional groups that would be needed to support patients are clearly limited:

“Then, of course, there must also be a non-physician, a non-biologist, a nutritionist or a dietician who then implements this, so to speak. And here, too, we see a big problem in medicine, that it is precisely in such non-physician professions where significant cost-cutting measures happen.”(Interview 53, clinician)

Another aspect experts talked about is the fact that patients sometimes have to overcome some barriers to even get access to a specialized inflammation outpatient clinic where PM is being researched. For example, a referral from a general practitioner or specialist is required for treatment at the clinic, which is sometimes withheld from patients:

“And there are still dermatologists who don’t refer patients to us even when they themselves have reached the limits of what else they can do themselves. So, it’s also amazing that patients repeatedly report: “I also said that I would like to get a second opinion somewhere else.” And then some of them are totally subhuman and virtually forbid the patients to do so. (…) I have already had the case that the specialist did not play along and a patient really had to expend a lot of energy to somehow come to us.”(Interview 59, clinician)

It is also reported that people with an immigration history are less likely to find their way to a specialized clinic. Additionally, long travel distances can be an obstacle. These admission barriers carry the risk that certain groups will be excluded from the benefits of PM. Furthermore, one expert emphasized that, for various reasons (time pressure, fear of regress, no specialization possible), it is also not to be expected that PM will become established in the outpatient sector, which would be easier for patients to attend. The respondents mentioned several factors that facilitate access. These factors included a patient being familiar with digital media, having fluency in the language (in our case, German) or belonging to a relatively well-educated group. According to the respondents, such patients are more likely to visit an inflammation clinic and will thus have easier access to PM in the future. 

### 3.2. Expectations for Future of PM

Although the development of PM for chronic inflammatory diseases is still in the research phase, the experts associate it with numerous hopes for the future. In the interviews, they describe the positive attributes and benefits of PM and what they expect as a positive outcome for patients.

#### 3.2.1. Attributes and Benefits of PM

The respondents expect multiple benefits from PM for the prevention and therapy of CID. Some experts labeled PM with positive terms like holistic care or medicine of the future (Table 4). 

Most of these benefits shown in Table 3 are likely self-explanatory, but others require clarification. For example, two professionals assumed that the communication between doctor and patient will improve as a result of the extensive data collection via wearables and smartphone apps. By using these devices, data are already available to the doctor before the consultation. This means that more time can be given to the patient discussion:

“What I wanted to say was that through precision medicine and, for example, also through apps that are given to patients, a much larger amount of data is possibly also made available to the doctor in advance. Otherwise, the doctor has to laboriously filter it out from a type of patient discussion. And usually in very poor data quality, because when I ask the patient how they have been in the last eight weeks, in most cases it is actually very much influenced by how the last two days were. That means that if it is used well, the opposite can actually happen when I have a very, very good picture beforehand—in terms of measurable disease parameters, subjective ones, the patient reported outcomes, and objective, quantitatively measurable ones. But then I really also have the time for the interaction to say: “Mrs. So-and-so, I have the impression here that in the last few weeks there actually—that you are not doing so well. What is the problem?” And [I] can actually be much more focused on what the patient actually needs.”(Interview 55, clinician)

One respondent noted that by collecting symptom and lifestyle data with wearables and smartphone apps, the patient also has the opportunity to learn which behavior is beneficial for them:

“And afterwards, the patient also learns more about what would be good for them and what needs to be done—if you also collect data, such as stress and all that and also really the fitness data—there is also sleep data (…), so that the patient can also see for themself: ‘Yes, okay, now I have a stressful phase here somehow, now I feel worse. You should somehow take a step back and somehow reduce the stress.’ Something like that.”(Interview 58, researcher)

Another expert believed that the data offer the opportunity to focus more on the patients’ individuality.

“Perhaps it is even more the case that one looks at the patient in such a way that every patient is unique and every patient is different. That you, as a doctor, have to put yourself in the individual’s shoes again and penetrate this jumble of data, so to speak, but see the patient as an individual. So, it is actually more of an opportunity, as I said, to improve the relationship between doctor and patient again.”(Interview 63, researcher)

Despite this positive assessment, three clinicians also perceived the risk that doctors then would focus too much on the data and give too little space to the patient’s narrative. However, one clinician explained that this is an existing problem: 

“This is a problem that already exists. Not only through omics. Today we have results from endoscopy, results from the laboratory, results from the CT. And sometimes you get the impression that the young colleagues do most of their medical work on the PC and no longer go to the patients. So, this is actually a general problem in medicine, which in principle will perhaps become worse through omics technologies. But I don’t think it will come about as a result. The problem is already there now.”(Interview 53, clinician)

During the validation workshop, one participant emphasized the benefit of precision medicine in terms of health care financing:

“One of the most convincing arguments in favor of precision medicine is that we may only be able to afford medicine in the future, in view of the increasing population age and thus the increasing morbidity of societies, we will no longer be able to afford the, let’s say, very lavish broad-spectrum medicine that we have at the moment. This means that in addition to the increase in well-being, lifetime, quality or whatever for the individual patient, it is also a matter of keeping medicine affordable at all in view of the demographic development.”(Validation workshop, researcher)

In addition to these predominantly positive expectations for precision medicine several experts emphasized that even with PM, there will be no cure, but only a remission of chronic inflammatory diseases. One expert highlighted that the benefits still must be proven scientifically. 

#### 3.2.2. A Good Life with CID

The experts are already pursuing the goal of improving or even restoring their patients’ quality of life. It also seems that biologicals are making a significant contribution to this goal compared to the drugs commonly used in the past. Many patients who could no longer imagine experiencing an improvement enjoy a significant increase in their life quality as a result of these drugs. One expert described that illustratively:

“They [patients] are sometimes quite surprised when they find their way to the doctor again and their physician says: ‘There is something [a new medication] new for you!’ And suddenly [after the patient has tried the new therapy] the psoriasis or neurodermatitis is under control, which had never been under control for 30 years and the patient has regained a life quality that was beyond their dreams. Today we have therapy options where patients really do have stable skin, findings that are almost completely free of symptoms and they almost forget that they once had the disease. (…) And I think that’s absolutely relevant, for example, that we always realize that we are in a position to give patients a good quality of life in the long term and to completely reintegrate them into normal life.”(Interview 62, clinician)

To measure the quality of life of their patients, clinicians use different patient reported outcome scales, for example the Dermatology Life Quality Index or the Short Form (36) Health Survey. However, these scales are not able to capture all of the dimensions of a good life, as one respondent clearly pointed out: 

“I can only say that we have devoted too little attention to this concept of a good life, or, as I said, the WHO’s technical term, well-being. We have limited ourselves to another standard of health-related quality of life. But this does not reflect the overarching concept of the good life or well-being. And we now have to start collecting data on this concept of a good life from patients in the first place. And that is actually something new for us.”(Interview 50, clinician)

When asking the experts about their ideas of a good life with CID, one receives a differentiated picture of what a good life with CID could be. Table 5 shows the subcategories developed from the experts’ answers. The most frequently mentioned aspects were that the disease does not affect the activities of daily living and that patients experience a remission of symptoms. While these aspects represent more of a health-related quality of life, some other factors go beyond this.

Implementing PM, however, could make a good life for patients even more likely because, as mentioned above, the tedious process of finding the right therapy is not a matter of trial and error, which is time consuming, a burden for the patient and cost intensive. The following quote from a clinician illustrates this:

“What we have been doing so far is that we give patients this antibody. We start with the first one, and after a defined period of time, which is about three months, we see whether this antibody has had an effect. And if it has worked, then we continue with the therapy. If it hasn’t worked, then we switch to the next one. Then the whole thing starts all over again. Three months of therapy and then we start evaluating again. (…) That’s where we are now. That wasn’t really precise. The first problem is that we have a lot of trial-and-error, which means that in extreme cases patients lose a lot of time until they actually reach a clinical remission—when the disease is really under control. They lose money because these drugs are expensive and if they are given them even though they don’t work at all, the money is practically just tossed in the wind.”(Interview 55, clinician)

## 4. Discussion

Our study describes the development of PM in chronic inflammation and, in particular, the collaboration made between professionals’ work and patients’ work in the context of the German research and health care system. We placed special focus on the ethical challenges that arise in the development of a valid database, patient education and therapeutic decision making, and in patient work. We discuss these challenges on the basis of different normative concepts. We start with the concept of justice, a principle that has been important since ancient times, but which is also discussed by contemporary philosophers [20,21]. 

Our research shows that several experts see the principle of justice violated by the fact that the data basis for PM development is predominantly limited to population groups with a European ancestry. This “epistemological bias” [11] may have the effect of excluding certain population groups from the benefits of PM, thereby exacerbating inequalities in health care. Korngiebel et al. provide some examples of the implications of not involving minorities in research: the authors point out that women of non-European genetic ancestry are more likely to receive a test result with a variant of unknown clinical significance when tested for hereditary breast and ovarian cancer than women of (traditional) European descent [22]. However, such disadvantages do not only affect different ethnic groups, as Lee shows. In her opinion, people with disabilities, representatives of the LGBTQ community, undocumented or uninsured people should also be included in the PM data base, as social factors like geography, income and health care access are important determinants of health [11]. Precisely these social factors were mentioned to some extent by the experts interviewed in our study as barriers for access to the inflammation clinic. In addition, disadvantages for women, which are significant in clinical trial inclusion [23,24,25] as well as in the prescribing of drugs [26], should be avoided in PM. To achieve inclusive PM, the possibility of a gender bias, caused by different behaviors or activities of men and women due to sociocultural norms, should be adequately addressed through the inclusion of gender in preclinical and clinical studies [25]. If a health care system is to be fair, then these barriers must be overcome, even if capacity limits and PM costs make this difficult to manage. Those facing such disadvantages should have facilitated access through active interventions in order to ensure that their chances of receiving high quality treatment available to the more fortunate. Further research is needed to determine which interventions can best overcome the barriers. 

Once precision medicine is applied, patient education and therapeutic decision making will also be of high importance. Additionally, as demonstrated in the interviews, it is not only medication that will play a role in a precise therapy concept. The patients’ work on their own health, for example by making dietary/nutritional changes that could also have a preventive effect, is also key. Especially when it comes to weight loss, many patients have difficulties in reaching their goals. Since being overweight is the result of many factors, including ethnicity, social class, neighborhood and other demographic factors [27], the stated need by one of our experts for support from dieticians, physiotherapists and psychologists is comprehensible. Political measures, such as the introduction of a sugar tax or traffic light labeling, as one expert suggested, could support these efforts and could be introduced in a cost-neutral way for the health care system. All of these interventions would be in line with the normative concept of an ethics of care, a practice of attentiveness to the needs of others—including their suffering, vulnerability and welfare. Not leaving others to their own devices and not letting them struggle alone are acts of humanity that the ethics of care commit to [28]. If PM continues to be applied and patient work becomes even more important, better solutions will have to be found for supporting patients in their efforts to adopt a healthy lifestyle. To this end, further research is needed.

The results of our study show that experts have different views on the future of SDM in PM. While one respondent believes shared decision making will become challenging in the context of PM because the molecular foundations of treatment decisions are difficult to understand, other clinicians are certain that SDM will still be possible. These clinicians emphasize that patient education will be less about explaining the molecular mechanisms of the disease. Rather, it will focus more on outlining which biomarkers support which therapy—and what therapy means for patients’ lives. The latter understanding of SDM corresponds rather to a “logic of care”, which the philosopher Annemarie Mol defines in contrast to a “logic of choice” [29]. Mol starts from the assumption that even in western societies, where autonomy is highly valued and people make decisions on their own, people still need support in therapeutic decision making. She argues that it is not enough to just present the therapy options, as would be the case with the common logic of choice. Rather, therapy decisions should be made together with the patient to see how this decision fits into the patient’s life [29]. Olthuis et al. also refer to this type of decision making as “caring decision-making”, emphasizing that therapy decisions involve consideration of the extent to which treatment affects patients’ well-being, values and plans for the future [30]. If patients are partly overwhelmed by SDM, as some of the experts report, then education in the course of the SDM and PM should focus more on topics that patients can easily understand. These are, in any case, patients’ values, plans for the future and their everyday lives. Balancing these areas with possible treatment options should be the goal of SDM in PM.

Another important principle of medical ethics has been the obligation not to harm patients [31], and this obligation must of course be followed further by modern PM. Protection against data misuse, discrimination and stigmatization, which was a concern in our study, but also in previous research with patients [32,33] and professionals [34,35], must be given high priority. To this end, technical solutions for data security at the interfaces between wearables, smartphones and EHRs must be developed and implemented so that lifestyle data can be easily integrated. The risk of stigmatization through the use of disease-specific health apps will likely not be completely eliminated. However, it can be mitigated if these apps do not refer to the disease in their app names. Since a residual risk of data misuse and possible discrimination also cannot be excluded, patients must be informed about this risk when they provide genetic and other data. A nationally uniform broad consent for the use in all hospitals, as developed by the German Medical Informatics Initiative [36], can facilitate this information and is an important milestone in expanding the PM database.

## 5. Limitations

As can be seen in Figure 1, our study was conducted at a time when the experts’ research remains in the process of building a valid database and identifying patterns in the data. Challenges later in the process were anticipated by the respondents based on their clinical and personal experiences. However, it is possible that these challenges may be perceived differently if precision medicine is actually applied more widely or if contextual factors, such as regulatory requirements, change. Future research on the ethical and social implications of precision medicine in chronic inflammation will therefore continue to be necessary.

## 6. Conclusions

With this study, we investigated the perspectives of researchers and clinicians in using PM to treat chronic inflammation. Thereby, we learned not only something about the current state of the development in this area, but also how these researchers and clinicians who are developing novel approaches of PM in CID conceive their own work processes of PM. PM is a distinguished style of research in biomedicine that uniquely combines professionals’ and patients’ work. Professionals also saw a series of ethical, process-related and economic challenges. This knowledge about the expected benefits, the particular research procedures that characterize PM and about the risks involved will enable interview studies with patients and their families. They will be necessary to understand the impact of PM on patients’ and families’ lives, to understand the value components in “patient related outcomes” from patients’ perspectives, and also to learn about patients’ and families’ needs to cope with the disease and its treatment.

Our research shows that questions of justice in precision medicine are a central ethical challenge. From our perspective, health care policy should therefore: (1) support precision medicine research that integrates diverse minority and ethnic groups to improve the data base for PM; (2) support research that explores new interventions to navigate disadvantaged people through the health care system and (3) support research and enact regulations that facilitate people’s healthy lifestyles, which is a particular difficulty for socioeconomically disadvantaged groups.

## Figures and Tables

**Figure 1 jpm-12-00574-f001:**
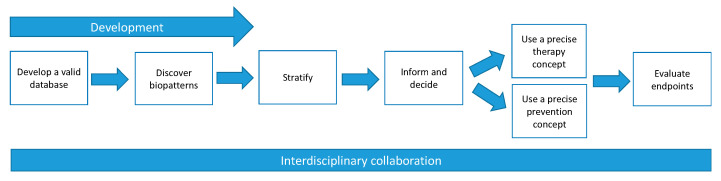
Professionals’ work in PM.

**Figure 2 jpm-12-00574-f002:**
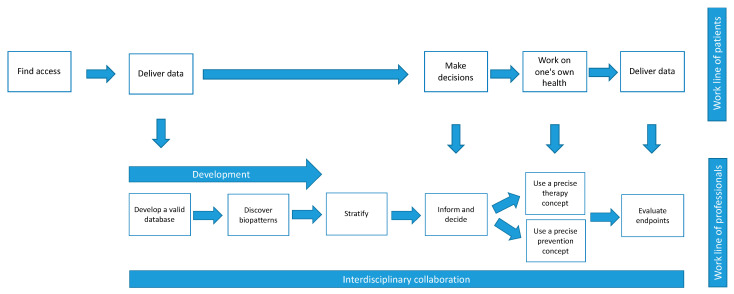
Patients’ and professionals’ work in PM.

**Table 1 jpm-12-00574-t001:** Participants’ characteristics.

Characteristics	Results
Age	
Mean (years)	45.4
Range (years)	30–66
Sex	
Men	12
Women	5
Profession	
Physician	10
Informatician/bio-informatician	2
Geneticist	2
Biologist	1
Interface designer	1
Registered nurse	1
Position	
Professor	11
Clinician scientist	2
Research assistant	2
Postdoctoral research fellow	1
Study nurse, nutrition consultant	1

**Table 2 jpm-12-00574-t002:** Open coding.

Main Category	Subcategory	Sub-Subcategory
Inflammatory diseases	Lung diseases	Asthma
		Chronic obstructive pulmonary disease
		Pneumonia
	Bowel diseases	Crohn’s disease
	Skin diseases	Psoriasis
		Atopic dermatitis
		Dermatological autoimmune diseases
	Rheumatological diseases	Psoriatic arthritis
		Rheumatoid arthritis
	Cardiovascular diseases	Arteriosclerosis
		Hypertension
	Metabolic diseases	Obesity
		Prediabetes
		Type 2 diabetes mellitus
	Neurological diseases	Migraine

**Table 3 jpm-12-00574-t003:** Positive Evaluation of the Broad Consent (BC).

Subcodes	Number of Codings
Different forms of consent fail to achieve their goals	1
Discarding of rest materials ethically not acceptable	2
Rollout of BC is required	1
No problem if data access is secured	1
Different data can be linked	1
Enables many new insights	2
Large amounts of data required	2
BC required in addition to targeted informed consent	1
Already practiced in other countries for a long time	1
High acceptance among patients	4
Requires no additional effort	4
Inhibiting data has negative consequences	1
Is the basis for PM	1
Benefits patients	2

**Table 4 jpm-12-00574-t004:** Attributes and Benefits of PM-Subcodes.

Subcodes	Number of Codings
No cure, only remission	7
Fewer side effects	7
Cost reduction	6
Holistic care	5
Improved efficacy	4
Improved doctor–patient communication	3
Improved quality of life	3
Enhanced safety	3
The patient learns what is good for them	3
Improved prediction of the course of the disease	2
Facilitates therapy decisions	2
Less medical care required	2
Improved compliance	2
Time saving	2
Moving beyond the principle of trial and error	2
Earlier treatment options	1
Medicine of the future	1
Lower burden for the patient	1
No sudden complications	1
Improved counselling for patients	1
Benefit still has to be proven	1

**Table 5 jpm-12-00574-t005:** What is a Good Life with CID?

Subcodes	Number of Codings
Disease does not affect everyday life	11
Symptom remission	10
Quality of life	4
Workability	4
Not feeling ashamed	3
Social life	3
Everybody judges this differently	3
Suppress inflammation	2
Statistically expected lifetime	1
Traveling	1
Pursuing hobbies	1
Be able to raise a family	1
Be glad to have received a good therapy	1
Have a competent professional contact	1
Patient realizes what they can do by themselves	1
Satisfaction of the physician	1
Patient satisfaction	1
Need to take as little medication as possible	1
Having accepted the disease as chronic	1
Not having to feel excluded	1
Joy in life	1

## Data Availability

The data are not publicly available due to ethical reasons. Public availability of data was not part of the informed consent we obtained from the participants.

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
