# Peer review of "Clinicians’ and Researchers’ Views on Precision Medicine in Chronic Inflammation: Practices, Benefits and Challenges"

_jpm, 2022, doi:10.3390/jpm12040574_

Round 1

Reviewer 1 Report

This is a very excellent paper. It is very interesting and very well written.

Author Response

Dear reviewer,

thank you very much for reviewing our paper and for your encouraging feedback.    We are glad that you found the paper interesting and well written.

Kind regards

Anke Erdmann

Reviewer 2 Report

A positive therapy response is obtained in cancer patients using PM as some studies demonstrated in the case of gastric cancer (https://pubmed.ncbi.nlm.nih.gov/33747967/) or colorectal cancer (https://pubmed.ncbi.nlm.nih.gov/34441055/).

The number of experts is very small, but the opinions are representative and their perspectives on the work processes of PM.

The figures are very helpful in understanding the assessed processes.

Author Response

Dear reviewer,

thank you very much for reviewing our paper and the helpful comments. We inserted a sentence into our introduction, outlining the positive therapy response for gastric cancer.

Kind regards

Anke Erdmann

Reviewer 3 Report

The authors explore the benefits and possible risks of precision medicine in chronic inflammation from the perspective of clinicians and researchers. The article is scientifically sound and well-written. I have only a few remarks that should be addressed.

Abstract:

  • "Ethical challenges were raised regarding the lack of integration of data from minority groups, the risk of data misuse and discrimination, the potential risk of no therapy being available for small strata, the lack of professional support and political measures in developing a healthy lifestyle, the problem of difficult access to the inflammation clinic for some populations and the difficulty of financing PM for all." --> Not all the challenges mentioned here are "ethical", especially the latter ones. The conclusions section correctly states "ethical, process-related and economic challenges".

Introduction:

  • "chronic inflammatory diseases (CID) such as ischemic heart disease, diabetes mellitus, stroke, cancer, chronic kidney disease, non-alcoholic fatty liver disease, autoimmune and neurodegenerative diseases are responsible for more than 50% of all deaths worldwide" --> Are these all considered CIDs here? What I understand from reference 1 is that inflammations could lead to these diseases. So they can be CID-related.

Results:

  • line 443: "likley" --> "likely"

Discussion:

  • Epistemological bias is discussed, but how about a gender bias? E.g. https://www.ahajournals.org/doi/full/10.1161/JAHA.119.014742

Author Response

Dear reviewer,

thank you for reviewing our article and the helpful comments and ideas. Especially the misunderstanding about chronic inflammatory diseases was very helpful and we changed this expression to "inflammation-related diseases" in the introduction. Your suggestion for considering a gender bias has prompted us to read some articles about this issue and we inserted two new sentences in the discussion. We are very grateful for this important thought. We also added "process-related and economic" to the "ethical challenges" in the abstract. All changes in the text are highlighted in red. 

Kind regards 

Anke Erdmann